# Hemi-Babim and Fenoterol as Potential Inhibitors of MPro and Papain-like Protease against SARS-CoV-2: An In-Silico Study

**DOI:** 10.3390/medicina58040515

**Published:** 2022-04-05

**Authors:** Ahmad Alzamami, Norah A. Alturki, Youssef Saeed Alghamdi, Shaban Ahmad, Saleh Alshamrani, Saeed A. Asiri, Mutaib M. Mashraqi

**Affiliations:** 1Clinical Laboratory Science Department, College of Applied Medical Science, Shaqra University, Al Quwayiyah 11961, Saudi Arabia; aalzamami@su.edu.sa; 2Clinical Laboratory Science Department, College of Applied Medical Science, King Saud University, Riyadh 11433, Saudi Arabia; noalturki@ksu.edu.sa; 3Department of Biology, Turabah University College, Taif University, P.O. Box 11099, Taif 21944, Saudi Arabia; ysghamdi@tu.edu.sa; 4Department of Computer Science, Jamia Millia Islamia, New Delhi 110019, India; shaban184343@st.jmi.ac.in; 5ICAR-Indian Agricultural Research Institute, New Dehli 110012, India; 6Department of Clinical Laboratory Sciences, College of Applied Medical Sciences, Najran University, Najran 61441, Saudi Arabia; saalshamrani@nu.edu.sa (S.A.); saaasiri@nu.edu.sa (S.A.A.)

**Keywords:** SARS-CoV-2, MPro, papain-like protease, molecular docking, molecular dynamics simulation

## Abstract

The coronaviruses belong to the Coronaviridae family, and one such member, severe acute respiratory syndrome coronavirus-2 (SARS-CoV-2), is causing significant destruction around the world in the form of a global pandemic. Although vaccines have been developed, their effectiveness and level of protection is still a major concern, even after emergency approval from the World Health Organisation (WHO). At the community level, no natural medicine is currently available as a cure. In this study, we screened the vast library from Drug Bank and identified Hemi-Babim and Fenoterol as agents that can work against SARS-CoV-2. Furthermore, we performed molecular dynamics (MD) simulation for both compounds with their respective proteins, providing evidence that the said drugs can work against the MPro and papain-like protease, which are the main drug targets. Inhibiting the action of these targets may lead to retaining the virus. Fenoterol is a beta-2 adrenergic agonist used for the symptomatic treatment of asthma as a bronchodilator and tocolytic. In this study, Hemi-Babim and Fenoterol showed good docking scores of −7.09 and −7.14, respectively, and performed well in molecular dynamics simulation studies. Re-purposing the above medications has huge potential, as their effects are already well-proven and under public utilisation for asthma-related problems. Hence, after the comprehensive pipeline of molecular docking, MMGBSA, and MD simulation studies, these drugs can be tested in-vivo for further human utilisation.

## 1. Introduction

A typical coronavirus (CoV), belonging to the Coronaviridae family, is identified by the crown-like spikes on its surface. The members of this family are numerous, and have caused various forms of destruction on several occasions, now in the form of the ongoing pandemic, which has led to millions of deaths worldwide [1,2,3]. The novel strain (nCoV) of severe acute respiratory syndrome coronavirus-2 (SARS-CoV2) causes mild to severe respiratory ailments, starting from symptoms such as fever, dry cough, sore throat, and conscious difficulties, and leading to death in the worst cases [4,5]. The World Health Organisation (WHO) declared COVID-19 a pandemic on 11 March 2020 [4]. The Worldometer reported 5,150,613 deaths out of 256,506,805 infected people worldwide by the end of 19 November 2021, with many more not counted because of the lack of proper information or medical support [6]. Most infected individuals experience mild to moderate symptoms, recover without treatment or quarantine, and are able to continue a healthy lifestyle. SARS-CoV-2 causes COVID-19, and it is transmitted through various modes, but the transmissive droplets are first generated after an infected individual coughs, sneezes, or exhales. Compared with past outbreaks of respiratory syndromes, such as Middle East Respiratory Syndrome (MERS) and SARS-CoV, the SARS-CoV2 (COVID-19) outbreak is considered the most worrying because of its infectivity rate and its methods of transmission. A low infection fatality rate (IFR) of 1.4% worldwide provides tremendous relief [7,8]. The first whole-genome sequence (WGS) of SARS-CoV-2 was made available to the public in January 2020, which has eased the job of researchers to develop diagnostic kits and understand the infectivity and severity rates [9]. Simultaneously, several labs have started providing the 3D structures of the proteins responsible for causing the primary infection and for binding with human angiotensin-converting enzyme 2 (ACE-2). This protein and genome information helped to carry out effective drug repurposing, alleviating the pandemic across the world by reducing the infectivity curve [10]. It is essential to understand how far we have succeeded in terms of medications for this pandemic, and why this virus has a high infection rate.

The world has already suffered a great deal, and the uncertainties have continued. There are no drugs available that can potentially target the virus. Scientists are designing or repurposing drugs against CoV-2 to reduce infections and fatalities [11]. This pandemic has demanded quick and effective healing strategies to combat the virus by reducing its dividing mechanism or making it inactive. Currently-approved drugs or compounds under clinical trials can be taken to reduce the process of approval if they can be utilised effectively against the disease. Moreover, they can be repurposed through combinational approaches of artificial intelligence (AI) and computer-aided drug design (CADD), leading to the identification of promising drug candidates in a short period of time. The present study focuses on regress screening and finding novel and approved drugs through molecular docking, MMGBSA filtering, and molecular dynamics simulation studies against the main protease and papain-like protease of SARS-CoV-2.

## 2. Methodology

We followed an extensive method of molecular docking (screening) analysis and binding free energy analysis. Additionally, we simulated the top-performing complex to establish stability and deviation. The graphical abstract is provided in Figure 1, and the description of the methods is as follows.

### 2.1. Protein Preparation

The 3D structures of SARS-CoV-2 main protease (MPro) and papain-like protease were downloaded from the RCSB (https://www.rcsb.org/, accessed on 15 January 2022) database with PDB id 6LU7 and 6W9C with no mutation at the resolutions of 2.16 Å and 2.70 Å, respectively. The X-ray-generated structures were not correctly configured, and the binding orientation needed to be fixed along with the hydrogen atoms. To fix all these issues and fill the gap, proteins were prepared using the ‘protein preparation wizard’ of Schrodinger Maestro (https://www.schrodinger.com/, accessed on 15 January 2022) (V.12.8.117) to fix all problems. The bond orders were assigned according to the CCD database, and the respective hydrogen atoms were added. The Prime module in the same wizard was used to fill the missing loops and side chains, and the hetero state was generated using Epik with a pH of 7.0 and zero bond orders to create the disulfide bonds [12]. In the papain-like protease, there were three chains, A, B, and C, but only chain A was kept, and all other heterodimers were removed, while in the case of MPro, chain A was also kept after review, and other solvents and ligands were removed. Furthermore, the H-bonds were optimised to fix all of the problems in the protein in the refine tab, and the OPLS_2005 forcefield was used for quickly restrained minimisation after removing all water molecules beyond 3.0 Å [13].

### 2.2. Ligand Library Collection and Preparation

A Drug Bank ID was created and approved by the Drug Bank team to access the data after agreeing to use it for only scientific purposes [14,15]. After logging in to the database, category-wise data were downloaded and further imported to maestro and grouped into one parent category. Furthermore, the LigPrep wizard was used to prepare the ligands. Some were initially in 2D format and not ready to dock, even though the hydrogen atoms did not satisfy the valency criteria [15,16]. We selected a pH of 7 (±2) to produce the possible states, kept only the best using Epik, and generated the tautomers. At most, 32 stereoisomers were kept for each ligand. 

### 2.3. Active Site Calculation and Glide Grid Generation

Using SiteMap, the active sites of the proteins (6LU7 and 6W9C) were determined [17]. During the computations, creating five active sites, the top-ranked probable receptor binding sites were identified, and crop site maps at 4 Å from the nearest site point were picked. The receptor grid was created using the receptor grid generation wizard. We determined the grid on site 1, increased the box to fit the entire active site, and performed molecular docking on the same grid [18].

### 2.4. Molecular Docking

A virtual screening workflow (VSW) was used to perform molecular docking, which combines all three methods to screen and score, i.e., HTVS (90%), SP (90%), and XP (100%). Further post-processing was performed with MMGBSA [19,20,21]. Using the QikProp tool, we filtered the ligands with their ADMET properties, refined them against Lipinski’s rule based on ADMET properties, regularised the input geometry, and deleted duplicates. Furthermore, the grid was added, and the job was kept by selecting all three algorithms [22]. 

### 2.5. Molecular Dynamics Simulation

After analysing the ligand interaction diagram, only one complex (protein–ligand) from each docking parameter was taken for molecular dynamics (MD) simulation. To explore the efficiency of the selected drugs by molecular docking, MD simulations were carried out using the Desmond package in Schrödinger suite v2021-3 [23]. Using the system builder in Schrödinger suite, the protein–ligand complex was prepared. After reducing the volume, the SPC water model was selected in orthorhombic shape with 10 Å × 10 Å × 10 Å periodic boundary conditions on the x, y, and z axes. Moreover, 3 Na^+^ were added to MPro, and 6 Cl^−^ were added to papain-like protease to neutralise the system. The exclusion of the ion and salt placement within 20 Å were considered.

Furthermore, the OPLS2005 forcefield minimised the energy of the complexes by heating and equilibrium processes before the simulations [13]. The complexes were treated with a steepest descent minimisation process before being heated at 0–300 K. In addition, the system was normalised in an equilibrium state at 1000 steps with a 100 ps time step. The system’s production step was extended up to 100 ns utilising the Nose–Hoover technique with an NPT ensemble, with a time step of 100 ps, temperature of 300 K, and pressure of 1.01325 atm [24].

## 3. Results

### 3.1. Ligand Library Preparation

The categorised drugs were imported and prepared with LigPrep, for a total of 155,888 ligands in different categories exported in a group (1 SDF file), and saved to a dedicated folder to use as a library for both conditions of the screening.

### 3.2. Molecular Docking

The virtual screening workflow resulted in 691 docked ligands that were further filtered on behalf of the docking and MMGBSA scores. The docking scores and their interacting residues were then analysed. Additionally, we only used protein–ligand pairs (papain-like protein–Hemi-Babim and MPro–Fenoterol) for further molecular dynamics simulation [25,26]. The protein–ligand interaction of the stable docked papain-like protein–Hemi-Babim and MPro–Fenoterol complexes was visualised with the ‘ligand interaction diagram’ to properly analyse the interacting residues. The Hemi-Babim (NH_2_, NH_2_^+^, and NH) showed two hydrogen bonds and one salt bridge with ASP179 and one hydrogen bond with HIS73, with a docking score of −7.090 (Figure 2A). The Fenoterol showed different bonding configurations: OH atoms (of Fenoterol) showed two hydrogen bonds with ASP197 and ASP289, and NH2+ showed two hydrogen bonds with LEU287 and ASP289 and one salt bridge with ASP289, with a docking score of −7.140 (Figure 2B). The repurposing of the above drugs has huge potential as their effects are already well-proven and under public utilisation for asthma-related problems. Our in silico screening through different algorithms of molecular docking studies and the prime MM-GBSA results show a promising output for confirming the compound’s activity and predict that the drugs can be further tested in vivo to understand their potent activity against SARS-CoV-2. The prime MM-GBSA results were used to calculate the binding free energy of the complex, which is also mentioned in Table 1. 

### 3.3. Molecular Dynamics Simulation

Molecular dynamics simulation is a method that utilises a set of algorithms to calculate and predict a compound’s stability. It is one of the best standalone mechanisms for a fundamental computational tool to capture the molecular and atomistic-level changes in and the stability of the protein–ligand complex (as determined by deviation and fluctuation studies) as well as for considering critical intermolecular interactions. Structure-based drug creation, employing traditional approaches such as molecular docking and virtual screening, has provided shortlisted drugs in bioscience. At the same time, MD simulation plays an essential role in understanding the ligand’s dynamic behaviour and its stability against the protein. For 100 ns molecular dynamics simulation in the SPC water model was kept for the production run. Further, the MD simulation trajectories were analysed with the Simulation Interaction Diagram (SID) to find the deviation, fluctuation, and intermolecular interaction. 

### 3.4. RMSD and RMSF

During the 100 ns MD simulation, the deviation in the protein’s backbone (C, C, and N) was calculated using the root mean square deviation (RMSD) value. As the temperature rises, the complex structure first swings and then stabilises. Both proteins did not deviate much during the entire simulation period. With regard to papain-like protease in complex with Hemi-Babim, the protein RMSD initially fluctuated from 0 to 1.17 Å in 1 ns (Figure 3A), while in the case of MPro in complex with Fenoterol the fluctuation in 1 ns was to 1.40 Å (Figure 3B). The total RMSD is acceptable in both combinations, and the fundamental fluctuations were due to the initial heat for the whole complex. The papain-like protease showed a deviation of 2.71 Å at 20 ns, and the MPro a deviation of 3.13 Å at 79 ns. Hemi-Babim and Fenoterol showed a higher deviation (red colour) because of rotatable bonds. The RMSD’s continued stability indicates that our compound was stable and well-bonded throughout the simulation period, resulting in an exemplary ligand interaction diagram.

The root mean square fluctuation (RMSF) analysis was used to find the complex fluctuations in time evolution. Figure 4A shows the protein-RMSF and the ligand contacts concerning each complex molecular dynamics simulation. We illustrated the papain-like protease and MPro in complex concerning ligands and their contacts to protein during 100 ns simulations. In papain-like protease, LYS190, GLN194, CYS226, and LYS315 showed the most fluctuation, and the rest of the residues showed a sig-nificantly less acceptable level of fluctuation. Hemi-Babim demonstrated 50 times contact with the protein, while among all the amino acids of MPro, GLY302, Phe305, and GLN306 showed the most fluctuation, with 73-time contact with Fenoterol during the complete 100 ns simulations. The noticed fluctuation was very low, and provides important information regarding the use of both drugs for further studies against CoV-2. Moreover, the intermolecular interactions analysis and secondary structure components, i.e., alpha helices and beta strands, make the protein molecule slightly rigid. The RMSF is shown in Figure 4A,B, in which can clearly be seen the fluctuations of less than 2 Å in all conditions, demonstrating promising results.

### 3.5. Intermolecular Interaction

During the simulation stage, atomic-level interaction studies are required to envision the ligands’ ability to bind to the protein. The intermolecular interactions between protein and ligand, such as ionic interactions, H-bonds, salt bridges, and hydrophobic contacts, were analysed for bond types during the simulative period of 100 ns. This intermolecular analysis confirmed many intramolecular interactions, including hydrogen bonds. In Figure 5A, we show the interaction of Hemi-Babim with the amino acids of papain-like protease and other relevant molecules. Although no direct interactions with carbon molecules were discovered, interactions with the NH, NH_2_, and NH_3+_ groups, which create H-bonds and hydrophobic and hydrophilic interactions with their corresponding percentiles, were observed. The arrow depicts the directions of donors and acceptors. The amino acids interacted directly and through the hydrophilic interactions, while water molecules interacted widely to form the water bridge. ASN128, ASP179, GLU295, THR158, ASP76, LEU178, and ASN156 formed hydrogen bonds with the shown respective percentiles during 100 ns simulations.

Figure 5B shows the interaction of Fenoterol with the amino acids of MPro. Interestingly, two different types of bonding were noticed: PHE3 formed pi–pi stacking with one benzene ring, while LYS5 formed a pi–cation interaction with the other benzene ring. Eight water molecules were involved in this interaction, and TRP207, ASP216, LEU282, GLU288, GLU290, THR199, ASN 214, ASP197, ASP289, and LEU287 formed hydrogen bonds with different atoms of the ligand. Furthermore, the statistical interpretations are provided in Figure 6A,B, showing the ionic interactions, hydrogen bond counts, hydrophobic interactions, and water bridges. 

## 4. Discussion

SARS-CoV-2 is accountable for millions of deaths globally and a high degree of morbidity due to its highly pathogenic nature. Today, many vaccines are available, but only for preventative use, and no curative drugs are known to date. There is a dire need for a remedial drug candidate to target the virus directly or indirectly by inhibiting nucleotide restrictions. We downloaded and prepared a complete library of the ligands from the Drug Bank database, which provides a vast amount of data, and we designed them to be ready to dock to the protein complexes [14]. Furthermore, two main targets, papain-like protease and, most importantly, the main protease, were targeted for the screening of the ligands, which gave tremendous results. Additionally, we used one complex from each set of docked results taken for the MD simulation and, interestingly, we found superior and stable results for both conditions. What fascinates us the most regarding the molecular docking results is the identification of Fenoterol (DB01288), a beta-2 adrenergic agonist bronchodilator that has already been used for different respiratory issues, and prominently so against asthma. The simulated ligands belong to a small group and show deviations and fluctuations less than 2 Å, with excellent ligand contacts. Fenoterol is an approved drug, while Hemi-Babim is an experimental categorised drug. However, an overdose of either drug is not suggested, as this can cause angina (chest pain), high blood pressure, muscle cramps, nausea, or a rapid heartbeat, and these drugs can only be used after prescription by medical professionals [25,26]. The mechanism of action of the proposed drugs is the stimulation of the beta (2)-receptor in the lung, which causes the relaxation of bronchial smooth muscle, bronchodilation, and increased bronchial airflow, which provide haptic ease to the patients. 

## 5. Conclusions

The reason for selecting the Drug Bank database was that it provides the drugs in a well-categorised manner, and, interestingly, the database is considered well annotated among all other drug databases. We applied three algorithms to reduce the computational cost, as the high-throughput virtual screening (HTVS) takes almost 1–2 s/ligand and only 10% of the best results are passed to the next step of the docking, standard precision (SP), and applied to the extra precise (XP) docking. Although our in-silico studies provided promising results with multiple ligands, we selected only the top ligand from both sets of results for further dynamic simulation using the SPC water model. The results for both candidates, i.e., Hemi-Babim and Fenoterol, showed the best scores in terms of docking, binding free energy (dG bind), and stable performance during the entire MD simulation period. The docked complex and binding free energy, as shown in Table 1, are of good enough quality to be relied on, as demonstrated by the ligand interaction diagram. The same was verified after MD simulation analysis for both protein–ligand complexes, after reviewing all outputs from the docking findings, binding free energy, and MD simulation. We believe that the selected medications may accommodate the papain-like protease and major protease pocket and prevent the infection mechanism of SARS-CoV-2. The potency of the studied compounds requires further investigation using experimental methods both in vitro and in vivo.

## Figures and Tables

**Figure 1 medicina-58-00515-f001:**
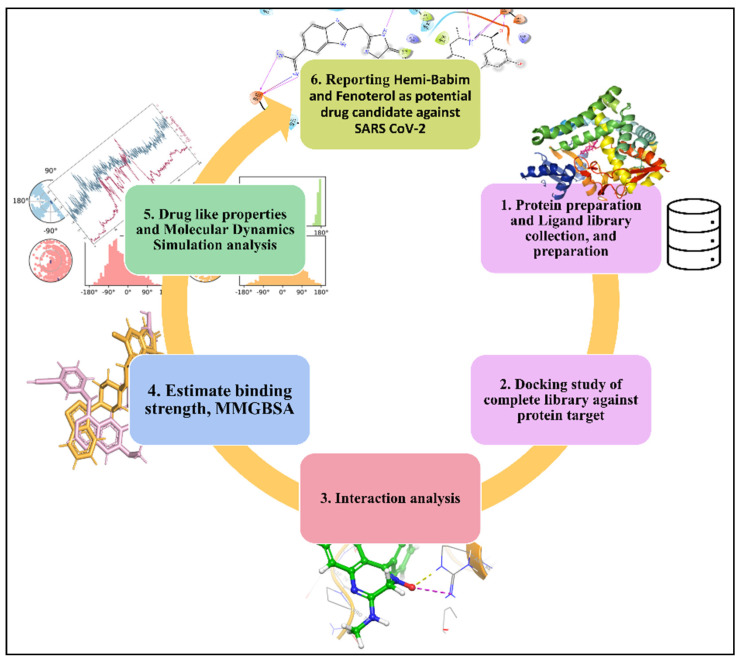
A graphical abstract of the study, showing the workflow from protein preparation and drug library collection to molecular dynamics simulation.

**Figure 2 medicina-58-00515-f002:**
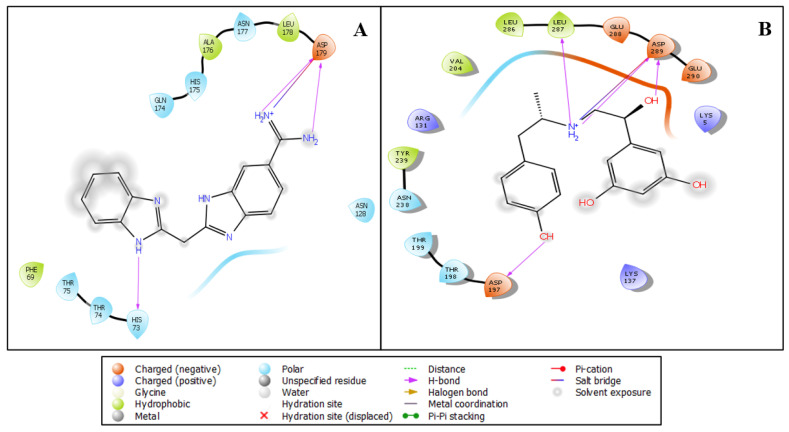
Ligand interaction diagram of (**A**) the papain-like protease and Hemi-Babim complex and (**B**) the MPro and Fenoterol complex, showing interacting residues and interaction types.

**Figure 3 medicina-58-00515-f003:**
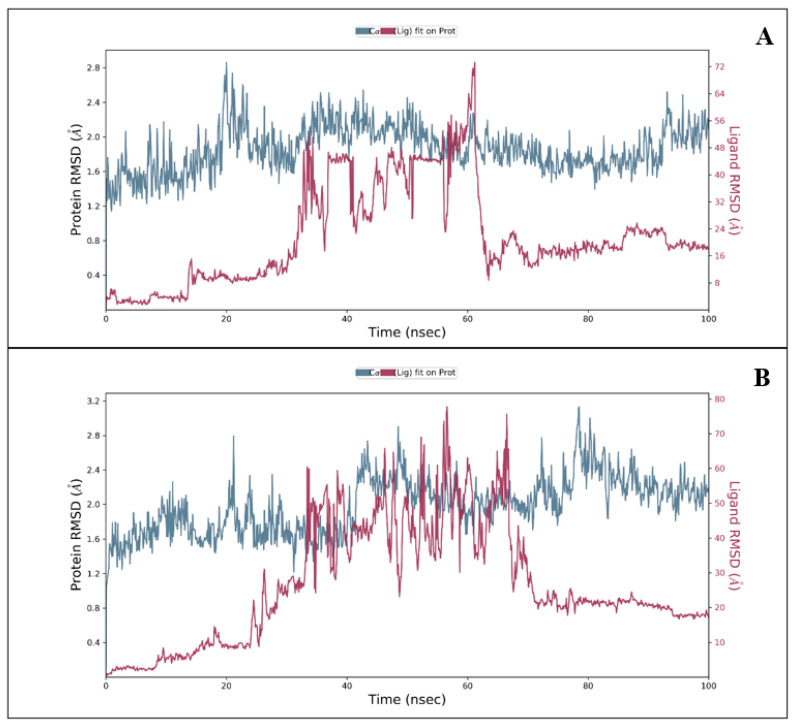
The RMSD for (**A**) the Hemi-Babim and papain-like protease complex, and (**B**) the Fenoterol and MPro complex.

**Figure 4 medicina-58-00515-f004:**
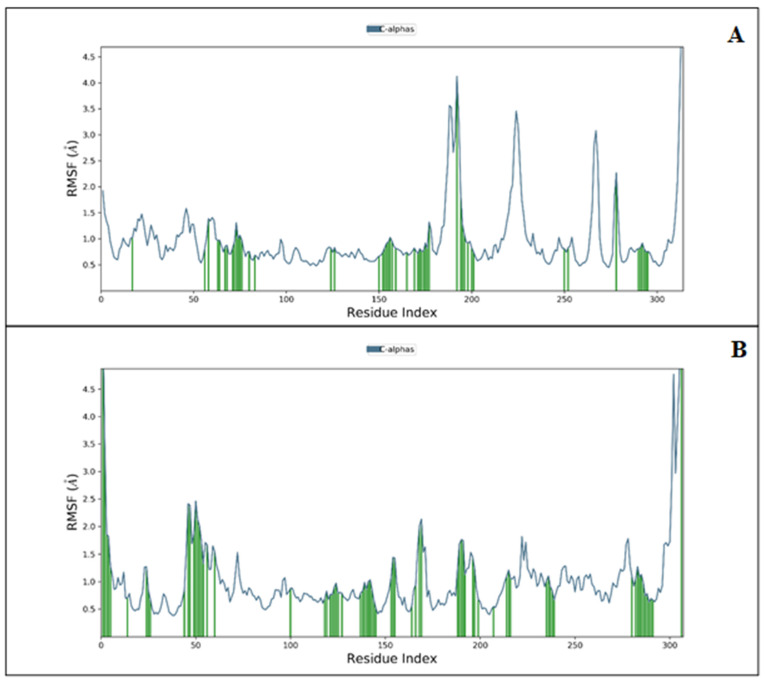
Protein-RMSF (blue) for (**A**) papain-like protease and (**B**) MPro concerning Hemi-Babim and Fenoterol (ligands contact—green).

**Figure 5 medicina-58-00515-f005:**
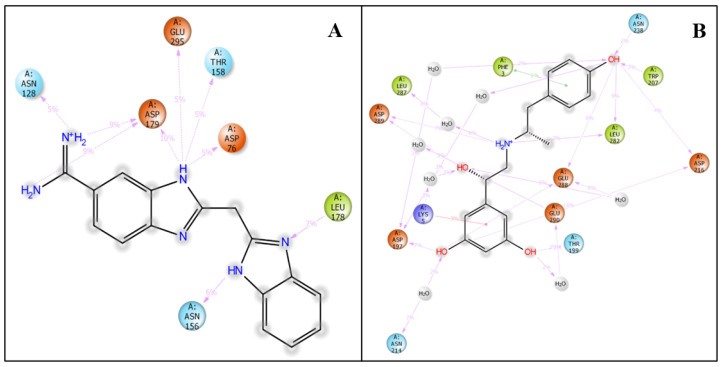
The 2D summary of interacting atoms with (**A**) the Hemi-Babim and papain-like protease complex and (**B**) the Fenoterol and MPro complex during 100 ns simulation.

**Figure 6 medicina-58-00515-f006:**
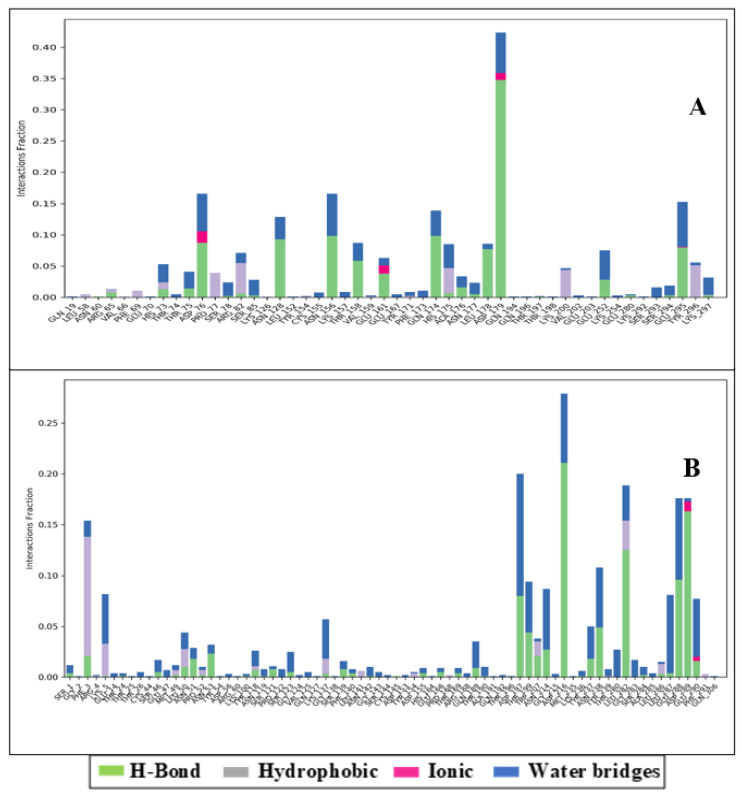
The interaction counts in histogram form for (**A**) the Hemi-Babim and papain-like protease complex and (**B**) the Fenoterol and MPro complex.

**Table 1 medicina-58-00515-t001:** The docking score and binding free energy of the Hemi-Babim and Fenoterol (Drug Bank ID), along with their respective proteins.

S. No.	Drug Bank ID	Protein Name	Drug	Docking Score	MMGBSA dG Bind	Rotatable Bonds	Ligand Efficiency sa	Ligand Efficiency ln	Evdw	Ecoul
1	DB01767	papain-like-protease	Hemi-babim	−7.09	62.392	3	−2.225	−7.977	−17.444	−71.398
2	DB01288	MPro	Fenoterol	−7.14	38.733	10	−2.812	−10.081	−26.532	−37.128

## Data Availability

All the data can be made public after publication.

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
