# Peer review of "Hemi-Babim and Fenoterol as Potential Inhibitors of MPro and Papain-like Protease against SARS-CoV-2: An In-Silico Study"

_medicina, 2022, doi:10.3390/medicina58040515_

Round 1
Reviewer 1 Report
Review report for:
Hemi-Babim and Fenoterol as potential inhibitors of MPro and papain-Like-protease against SARS CoV-2: An in-silico study
In this paper, the authors screen ligands from the drug databank to find a possible drug for SARS-CoV-2.
In general, I think the paper is interesting and should be published. However, the English has to be improved significantly. In addition, the authors should be careful with small errors in their MS. For example:
Page 1, line 19: there are two “is” and the “(” has to be closed.
Page 1, line 22: “(“ has to be closed
Additional notes:
Method section:
Page 4: line 127: the terms of the equation should be explained clearly.
I don’t understand this statement: “After analyzing the ligand interaction diagram, only one complex (protein-ligand) from each docking parameter was taken for the molecular dynamics (MD) simulation.“
Results Section:
The numbers in table 1 (vdw and Coul.) are merged. Please fix this.
Figure 4 A. The figure is not shown appropriately (the authors have to be careful with these issues)
For the intermolecular interactions, it would be very useful to have a figure that shows the total interaction energy between the protein and the ligand through the full MD simulations. This would be useful to identify stability of the binding.
Reviewer 2 Report
Dear authors,I appreciate your hard work to screen the
complete library from the Drug bank and its mining to MD simulation work. Your
manuscript is well written and can deliver something good for the other
researchers. Anyhow, I have a minor suggestion that I would like to see before
its acceptance. Comments-1: The MPro is mentioned for the main
protease, and it can be modified with the Main protease, and the abbreviation should
be Mpro. It needed to be modified throughout the journal.
Comment-2: Why the author has selected only
the main protease and Papain-like protease, not any other? Any specific reason behind
this?
Comment-3: The formula for the GideScore can
be removed if it is not necessary.
Comment-4: Not sure if the references match
with the journal requirements. If not, then it can be updated to the journal format.
I would be happy to see all the suggested updates
before the acceptance. I wish all the best to the authors.
